# Spatial Neglect in Stroke: Identification, Disease Process and Association with Outcome During Inpatient Rehabilitation

**DOI:** 10.3390/brainsci9120374

**Published:** 2019-12-13

**Authors:** Ulrike Hammerbeck, Matthew Gittins, Andy Vail, Lizz Paley, Sarah F Tyson, Audrey Bowen

**Affiliations:** 1Division of Neuroscience and Experimental Psychology, Faculty of Biology, Medicine and Health, University of Manchester, MAHSC, Manchester M13 9PL, UK; ulrike.hammerbeck@manchester.ac.uk; 2Centre for Biostatistics, Faculty of Biology, Medicine and Health, University of Manchester, MAHSC, Manchester M13 9PL, UK; Matthew.Gittins@manchester.ac.uk (M.G.); Andy.Vail@manchester.ac.uk (A.V.); 3School of Population Health and Environmental Sciences, Kings College London, London SE1 1UL, UK; Lizz.Paley@phe.gov.uk; 4Division of Nursing, Midwifery and Social Work, University of Manchester, MAHSC, Manchester M13 9PL, UK; sarah.tyson@manchester.ac.uk

**Keywords:** spatial neglect, stroke, severity, length of stay, dependency, outcomes, therapy, rehabilitation

## Abstract

We established spatial neglect prevalence, disease profile and amount of therapy that inpatient stroke survivors received, and outcomes at discharge using Sentinel Stroke National Audit Programme (SSNAP) data. We used data from 88,664 National Health Service (NHS) admissions in England, Wales and Northern Ireland (July 2013–July 2015), for stroke survivors still in hospital after 3 days with a completed baseline neglect National Institute for Health Stroke Scale (NIHSS) score. Thirty percent had neglect (NIHSS item 11 ≥ 1) and they were slightly older (78 years) than those without neglect (75 years). Neglect was observed more commonly in women (33 vs. 27%) and in individuals with a premorbid dependency (37 vs. 28%). Survivors of mild stroke were far less likely to present with neglect than those with severe stroke (4% vs. 84%). Those with neglect had a greatly increased length of stay (27 vs. 10 days). They received a comparable amount of average daily occupational and physiotherapy during their longer inpatient stay but on discharge a greater percentage of individuals with neglect were dependent on the modified Rankin scale (76 vs. 57%). Spatial neglect is common and associated with worse clinical outcomes. These results add to our understanding of neglect to inform clinical guidelines, service provision and priorities for future research.

## 1. Introduction

Spatial neglect is a distressing consequence of stroke [1] associated with a worse outcome [2]. It is a heterogeneous syndrome rather than a single impairment [3]. Central to neglect is a cognitive disorder of attention and awareness which manifests as an ipsilateral behavioural bias and “exaggerated spatial asymmetry in processing information”, whereby patients characteristically fail to orientate, report or respond to stimuli on the contralesional side [1,4,5]. Neglect is usually caused by large strokes in the middle cerebral artery territory and the manifestation of neglect tends to be more severe and persistent following right hemisphere damage [1,5]. Critically for rehabilitation, many neglect patients present with anosognosia and deny difficulties with perception or control of movement, being not aware that they are experiencing these symptoms [6,7]. Therefore, it is not surprising that neglect early after stroke [8,9], as well as enduring neglect [10,11,12], is a prognostic indicator for reduced functional independence following stroke [13]. This is further impacted by the association of neglect with more severe sensory impairment [14] which is in itself a prognostic indicator of poorer recovery after stroke [15].

Neglect hinders the ability to participate in therapy to improve functional independence [16,17], limiting recovery in the period of heightened neuroplasticity early after stroke [11,18,19]. It is an independent indicator of upper limb use and thereby predictor of recovery [20] with limited evidence of effective cognitive rehabilitation regimes [21]. Neglect impacts daily activity and can result in safety issues (e.g., falls) and decreased likelihood of living independently with a resultant reduction in psychological well-being [20,22,23,24]. 

Although it appears as though neglect is associated with greater age, greater number of co-morbidities and a worse premorbid health profile no generalisable large scale study has been performed to obtain a representation of neglect at a clinical practice level. It is plausible that the greater dependence observed in stroke survivors with spatial neglect results in longer hospital stay and increased use of resources as well as a greater burden on informal carers and formal support at discharge. 

We therefore analysed a two year database of ~150,000 consecutive stroke admissions in England, Wales and Northern Ireland to describe the population of stroke survivors with neglect in comparison to individuals with no neglect. We established the demographics, stroke severity, comorbidities, their resource use and clinical outcomes on discharge from in-patient care for the two groups. This information will be important to inform service provision as well as for the development of research protocols, and priorities for future research.

## 2. Materials and Methods

We extracted data from an anonymised dataset from the Sentinel Stroke National Audit Programme (SSNAP) [25,26]. SSNAP collects clinical, patient-level as well as acute and post-acute organisational-level data for all stroke patients admitted to hospital in England, Wales and Northern Ireland. For this study we used the clinical component, a longitudinal register of patient-level information including demographics, pre-morbid disability, comorbidities, stroke characteristics, in-patient treatment received, and health outcomes (disability on the modified Rankin Scale [27,28]) at discharge from inpatient care.

We extracted data for patients admitted to hospital across the three nations with a confirmed diagnosis of stroke between July 2013 and July 2015, if they had a completed baseline neglect assessment for the National Institute for Health Stroke Scale (NIHSS), see Table 1 [29]. We applied a further inclusion criterion that individuals were still in-patients at three days. Therefore, in our analysis of SSNAP we excluded individuals who died, went to palliative care or were discharged within three days of admission. Our rationale for these criteria was to focus on those most likely to be receiving specialist inpatient stroke care and rehabilitation beyond the hyperacute stage. 

## 3. Analysis

Presence of spatial neglect was indicated by a score > 0 on question 11 (extinction and inattention) of the NIHSS (Table 1). On admission to hospital, stroke impairment is routinely measured using the NIHSS [29] and documented in SSNAP. The NIHSS is a quick, simple, psychometrically robust (if relatively crude) measure of overall stroke severity which assesses the severity of stroke-related impairments. It contains 15 items which measure: level of consciousness (consciousness, orientation, ability to follow commands); cognition (language and extinction/inattention); vision (motor visual-field loss and extraocular movement); motor control (weakness of the limbs; ataxia, dysarthria), and sensory loss. A trained observer rates the patient’s ability to answer questions and perform activities. The presence and severity of impairments is rated and summed to a total score, with a maximum of 42. A score of zero indicates the absence of stroke symptoms and the higher the score, the more severe the stroke. Only one item of the NIHSS is mandatory in SSNAP (Level of Consciousness), so patients may have missing values for the remaining 14 items.

We used descriptive statistics to establish the prevalence of spatial neglect in this in-patient population of stroke survivors and describe the demographics and disease profile in the group with neglect and the group without neglect. For the baseline characteristic we present numbers and percentages to indicate the presence of neglect by specific factors (gender, stroke severity, premorbid disability, co-morbidities and stroke type). We then present numbers and percentages for those with and without neglect in respect of the amount of therapy received, length of in-patient stay and outcome/disability (modified Rankin Scale on discharge). Note statistical inferences have not been performed. Due to the large sample size, it would be highly likely that any simple comparison (e.g., t-test) would be statistically significant regardless of the size of effect. The data structure and confounding present in this observational, records-based dataset would require complex methodology to produce robust conclusions. Therefore, and given our objective to describe the sample population, these were not planned for a priori or reported here.

We extracted data on physiotherapy and occupational therapy and considered several options to quantify amount of therapy received from the routinely collected SSNAP data available to us. A simple ratio of minutes/days would produce the average therapy received per day on which patients received treatment, however patients rarely received therapy every day therefore, to more accurately represent their therapy amount over the care period, and to limit the influence of any self-reporting bias, we used ‘average therapy per day of stay as an inpatient’. Due to the nature of the therapy variables collected in SSNAP, information on which days during the in-patient stay therapy was received is not recorded, limiting a more detailed analysis such as frequency (e.g., whether patients received all their minutes of therapy in a single session per day, if it was delivered by more than one therapist or spread across several shorter sessions).

We report the mean (standard deviation) where data were plausibly normally distributed and the median (interquartile range) for data where this is not the case.

## 4. Results

During the data extraction period (July 2013 to July 2015) 149,560 stroke patients were admitted to hospital and entered into the SSNAP clinical audit. 41,706 were excluded as they had a length of stay (LoS) <3 days (whether due to death or early discharge) or received palliative care. A further 18% of patients were excluded due to an incomplete neglect component of the NIHSS score. In 12,949 patients (12%) of these the ‘Level of Consciousness’ on admission was recorded as zero (i.e., ‘alert’) but all other NIHSS items were incomplete, and in 6241 patients (6%) the neglect assessment for the NIHSS was incomplete resulting in a final tally of 88,664 records for our analyses. 

The demographics of the sample are documented in Table 2. For this population we found that spatial neglect was identified and recorded on the NIHSS in 30% of admissions. Neglect was observed in a slightly older population (78 years) in comparison to the group without neglect (75 years). Neglect was more commonly observed in females than in males (33% vs. 27%).

As stroke severity increased so did the presence of spatial neglect (mild = 4%, moderate = 32%, moderate to severe = 69%, severe = 84%). Pre-morbid dependency (mRS) was associated with a higher presentation of neglect (37% vs. 28%). The two comorbidities in which neglect was observed more commonly was for individuals with congestive heart failure (CHF) in comparison to those without CHF (34% vs. 30%) and atrial fibrillation (AF) in comparison to individuals without AF (38% vs. 28%). As expected, ischaemic stroke was far more common than haemorrhagic stroke in this sample, but neglect was detected more often in haemorrhagic stroke (36% vs. 30%). The frequency of co-occurrence of neglect with other impairments such as hemianopia and hemiplegia is shown in Appendix A. Here we also present data on missing data, i.e., the total sample of 94,905. 

Table 3 reports how frequently the need for physiotherapy (PT) and occupational therapy (OT) was documented and whether patients received therapy if it was indicated. We only included this information for the initial admission. Patients with neglect were marginally more likely to require physiotherapy than those without neglect (neglect 94% vs. non neglect 92%) but marginally less likely to require OT compared to patients without neglect (87% vs. 90%). The median minutes of physiotherapy received per day of inpatient stay, for those individuals who required it, was very similar (13.8 min) for neglect compared to those without (13.2 min), but slightly fewer minutes of OT per day of inpatient stay were received by those with neglect (11.3 min) compared to individuals without (12.7 min).

Individuals with spatial neglect, excluding those who died during hospitalisation, had more than double the length of inpatient stay (LoS) (27 days), than surviving patients without neglect (10 days). At discharge, dependency on others or death, indicated by a modified Rankin Scale score of greater than two, was much greater in patients with spatial neglect (76%) in comparison to those without neglect (57%) (Table 4). Mortality during hospitalisation (but after 3 days) was much higher in individuals with spatial neglect (23%) compared to patients without neglect (8%). For the survivors the discharge destination from hospital was different for individuals with spatial neglect, in that 26% of these individuals were discharged to a care home compared to 12% of those without neglect, and fewer people with neglect returned to their own/family home on discharge (53%) compared to individuals without neglect (72%).

Referral to ongoing therapy provision did however not differ greatly between individuals with and without neglect. Twenty-four percent of individuals with neglect, and 26% of individuals without neglect were referred to both early supported discharge (ESD) services as well as a community rehabilitation team (CRT). ESD referral without referral to CRT was documented in 9% of individuals both with and without neglect, and CRT without referral to ESD in 20% of individuals with, and 23% of individuals without, neglect. No onward referral on discharge was documented in 45% of individuals with and 44% of individuals without neglect. 

## 5. Discussion

Our observational study, of national stroke registry (SSNAP) data over a two-year period of 88,664 admissions to hospital with stroke in England, Wales and Northern Ireland, provides a unique opportunity to understand neglect in a real-world acute hospital population. We found that in this patient population neglect occurs in at least 30% of stroke survivors. Neglect was associated with an older age and with greater pre-stroke levels of dependency. Furthermore, neglect is associated with a more severe stroke, increased length of hospital stay and greater disability and dependence on discharge, increased mortality and likely discharge to a care home. The large sample size makes these findings widely generalizable, and presents a clinical picture of a frail, vulnerable and sizeable subpopulation in likely need of long term health and social care. Given the absence of strong evidence for effective neglect rehabilitation [21] there is an obvious need for the development and robust evaluation of the clinical and cost effectiveness of strategies to identify and rehabilitate people with neglect.

When investigating the association of age and neglect, previous studies have reported that it is prevalent in an older generation [30] which our data support, as the group with neglect was on average three years older. An unexpected finding was that neglect was observed more frequently in women, as previous studies found the incidence to be equal across genders when performing drawing tasks [31]. As previously reported, we confirm that neglect is associated with a more severe stroke [30,32] but the extent of this is very marked in our data, ranging from 4% of those with mild strokes through to 84% of those with severe strokes. We found that neglect was more common in individuals who were dependent before their admission, but co-morbidities such as diabetes and previous stroke/Transient Ischemic Attach (TIA) although present were curiously less frequent. It is unclear why that is although more people with neglect died (23% vs. 8%) during the inpatient phase that we studied (alive and still in-patients after three days). Alternatively, there may be selection bias in entering routinely collected clinical data into the national audit database with clinicians and audit clerks more focused on severe stroke related deficits than milder problems in the person’s medical history. 

Therapy effectiveness is reduced in stroke survivors with neglect [33] because neglect hinders patients’ ability to participate in functional therapy [16,17], with a detrimental effect on upper limb use [18] and recovery as a whole [20]. We were therefore interested to evaluate therapists’ perception of the therapy need of this population. Interestingly the greater severity of the stroke did not appear to greatly influence therapists’ perception of whether the individuals required therapy, nor the amount of daily therapy that individuals with neglect received. It is interesting to note, but difficult to explain, why people with neglect were less likely to be considered for occupational therapy than physiotherapy (87% vs. 94%), although this may be a resource issue as staffing levels in the UK are lower for occupational than physiotherapy [26]. It is a concern that, for all stroke survivors in this study, the average of less than 14 minutes of physiotherapy and even less of occupational therapy per day of inpatient stay, is far less than they are likely to need and as recommended in national clinical guidelines [34].

We found that length of hospital stay was more than double in individuals with spatial neglect and neglect was associated with much greater dependency at discharge and death. This is in line with previous reports of the poor longer term outcome of stroke survivors with neglect [13]. People with neglect were more likely to be discharged to a care home, probably an indication of their increased dependency. There is a clear economic as well as a human cost to these findings. 

It is encouraging to see that onward referral for specialist rehabilitation (early supported discharge or community rehabilitation) is just as likely for people with neglect, but we do not have further data on the amount or type of therapy they receive nor on the outcomes. Although it is widely accepted that some recovery occurs over time [35] the prevalence and functional impact of spatial neglect in the chronic period post-stroke remains unclear as there is no recent, adequately–sized, representative study of the long term prevalence of spatial neglect.

## 6. Strengths and Limitations

The main strength of this analysis is the large size of the data-set consisting of all consecutive admissions in England, Wales and Northern Ireland between July 2013 and June 2015. These findings are therefore a true reflection of the prevalence and outcome of neglect as it is currently assessed in the national clinical services. 

For our analysis, the presence of neglect was indicated by the National Institute for Health Stroke Scale (NIHSS). Neglect is reported as (0) no abnormality detected, (1) visual, tactile, auditory, spatial or personal inattention or (2) profound hemi-inattention or extinction to more than one modality. We recognise the simplification, limitation and imprecision of this measure and that it is possible that mild or atypical impairments, like personal neglect [36], may have been missed with a likely underestimation of the prevalence of neglect [37,38]. In line with this assumption, the prevalence of neglect varies between studies, depending on the choice and timing of test used, diagnostic criteria and different types of neglect [39,40,41]. We further recognise that better, more detailed assessments are available, like the Oxford Cognitive Screen (OCS) [42] however, we performed analysis with the measure which is routinely performed in clinical practice for all stroke patient admissions in England, Wales and Northern Ireland and on balance feel that these data make a useful contribution. 

A weakness of using observational data is that causation cannot be inferred. Analysis is restricted to variables collected in SSNAP and further data validation was not possible. Therefore we cannot say that the poor outcomes observed were caused by the neglect and acknowledge that they could be attributable to another symptom, like hemiplegia or hemianopia, see Appendix A. Due to the nature of the dataset and the descriptive analysis we performed, we at no point statistically analyse any associations with outcome measures, and in turn would not be able to identify if causal relationships are present. Any interpretation should therefore be done with great care. 

For this size of database, it would be of great interest to record the incidence of neglect in left and right hemisphere stroke. Unfortunately, this is not information that is captured in the SSNAP database and therefore we cannot report on this aspect. SSNAP does contain NIHSS data on left and right arm and leg (Q5–6) however we cannot be certain that these motor impairments resulted from the recent stroke and so cannot use them to extrapolate side of lesion. The breakdown of the observed weakness is presented in Appendix A.

18% of patients in the whole database had a missing NIHSS and were therefore excluded in the analysis; it is therefore possible that our finding of a 30% prevalence of neglect is either and underestimation or an overestimation for the entire stroke population. The assessment for neglect is conducted on arrival in the very acute phase of stroke. As impairments resolve quickly in the acute stage, it could be that some of these resolve [32]. The data are also subject to data entry error as quality control for a national database is not feasible. 

## 7. Conclusions

Our analysis demonstrates that spatial neglect is common, seen in a frail population and associated with worse clinical and process outcomes. These results add to our understanding of neglect and can be used to inform clinical guidelines, service provision and priorities for future research. Greater understanding of the profile of individuals with neglect will assist in designing future trials.

## Figures and Tables

**Table 1 brainsci-09-00374-t001:** National Institute for Health Stroke Scale (NIHSS) Question 11.

Q11 NIHSS	Extinction and Inattention:
0	Normal; patient correctly answers all questions.
1	Inattention on one side in one modality; visual, tactile, auditory, or spatial; Extinction to bilateral simultaneous stimulation
2	Profound hemi-inattention. Hemi-inattention; does not recognize stimuli in more than one modality on the same side.

**Table 2 brainsci-09-00374-t002:** Descriptive statistics of the baseline demographics and stroke characteristics of individuals without and with neglect. All data are reported as the number of individuals and the row percentage of the descriptors resulting in individuals presenting without or with neglect.

Baseline	Measure	No Neglect *n* = 61,948 (70%)	Neglect *n* = 26,716 (30%)	Whole Sample *n* = 88,664
**Age**	Mean (S.D) Min: Max	75 (13.5) 18:114	78 (12.3) 18:114	75.8 (13.2) 18:114
**Gender**	Female Male	30,492 (67%) 31,456 (73%)	15,058 (33%) 11,658 (27%)	45,550 43,114
**NIHSS Score**	Med (IQR) Min: Max	4 (2:7) 1:42	15 (9:20) 0:42	6 (3:12) 0:42
**Stroke Severity on NIHSS**	Mild(<5) Moderate (5–14) Moderate-Severe (15–20) Severe (>20)	33,830 (96%) 23,789 (68%) 2810 (31%) 1519 (16%)	1590 (4%) 10,995 (32%) 6244 (69%) 7887 (84%)	35,420 34,784 9054 9406
**Pre-mRS**	Independent (≤2) Dependent (>2)	50,925 (72%) 11,023 (63%)	20,148 (28%) 6568 (37%)	71,073 17,591
**Co-Morbidity**	Congestive Heart Failure Hypertension Atrial Fibrillation Diabetes Previous Stroke/TIA	3418 (66%) 34,338 (70%) 12,123 (62%) 13,225 (73%) 17,318 (71%)	1801 (34%) 14,871 (30%) 7516 (38%) 4974 (27%) 7241 (29%)	5219 49,209 19,639 18,199 24,559
**Stroke type**	Infarction ICH	55,568 (70%) 5803 (64%)	23,358 (30%) 3206 (36%)	78,926 9009

IQR = inter quartile range, TIA = transient ischaemic attack, ICH = intracerebral haemorrhage.

**Table 3 brainsci-09-00374-t003:** Descriptive statistics of when physiotherapy (PT) and occupational therapy (OT) were documented to be required, and whether this was then provided. All data are reported as the number of individuals and the column percentage within the specific sample of individuals with and without neglect.

Therapy Descriptive			No Neglect *n* = 61,948	Neglect *n* = 26,716	Whole Sample *n* = 88,664
PT	Required at first entry		57,178 (92%)	25,048 (94%)	82,226
	If required, did they receive it?	No	435 (1%) 56,743 (99%)	196 (1%) 24,852 (99%)	631 81,595
		Yes			
OT	Required at first entry		55,584 (90%)	23,300 (87%)	78,884
	If required, did they receive it?	No	591 (1%) 54,993 (99%)	321 (1%) 22,979 (99%)	9191 77,972
		Yes			
PT per day of stay (minutes)		Median (IQR) Min:Max	13.2 (7.4, 21.2) 0:218	13.8 (7.5, 22.1) 0:244	13.3 (7.5, 21.4) 0:244
OT per day of stay (minutes)		Median (IQR) Min:Max	12.7 (6.8, 21.1) 0:229	11.3 (5.2, 20) 0:231	12.4 (6.3, 20.7) 0:231

**Table 4 brainsci-09-00374-t004:** Descriptive statistics of health outcomes for patients with spatial neglect in comparison to the whole population. All data are reported as the number of individuals and the column percentage within the specific sample of individuals with and without neglect.

	Health Outcomes Descriptive (Column %)	No Neglect *n* = 61,948	Neglect *n* = 26,716	Whole Sample *n* = 88,664
Length of stay for survivors in days	Median (IQR) Min:Max	10 (5, 26) 3:765	27 (10, 56) 3:804	13 (6, 34) 3:804
mRS outcome	Independent (≤2) Dependent or dead (>2)	31,493 (43%) 30,455 (57%)	6413 (24%) 20,303 (76%)	37,906 50,758
Deaths as inpatient (after 3 days)		4877 (8%)	6254 (23%)	11,131
Survivors		57,071 (92%)	20,462 (77%)	77,533
Discharge destination from inpatient stay (% of survivors)	Care Home Home Unknown location died (with ESD/CRT) SSNAP No SSNAP Somewhere else Other Inpatient non-SSNAP	6738 (12%)	5306 (26%)	12,044
		40,977 (72%)	10,763 (53%)	51,740
		191 (<1%) 2323 (4%) 1276 (2%) 4036(7%)	132 (1%) 745 (4%) 520 (3%) 2166 (11%)	323 3068 1796 6202
		1530 (3%)	830 (4%)	2360
Therapy referral on discharge (% of survivors)	ESD and CRT ESD only CRT only neither	13,843 (24%) 5181 (9%) 13,042 (23%) 25,005 (44%)	5240 (26%) 1932 (9%) 4104 (20%) 9186 (45%)	19,083 7113 17,146 34,191

mRS = Modified Rankin scale, ESD = early supported discharge, CRT = community rehabilitation team.

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
