# Peer review of "Spatial Neglect in Stroke: Identification, Disease Process and Association with Outcome During Inpatient Rehabilitation"

_brainsci, 2019, doi:10.3390/brainsci9120374_

Round 1

Reviewer 1 Report

I read with interest this large scale - population based analysis. The paper is well written and reports an extensive analysis that confirm previous studies in a different form, analyses and population. The authors correctly conclude that the results are useful to take decision in future policies, statements and service provision. However there are some minor points to change to improve the readability of manuscript to a wider audience and include some recent points.

Title

Ok

Abstract

I think the “modified Rankin” is the “modified Rankin Scale.”. Please add this description.

Introduction

I agree with the description of spatial neglect, but a recent definition could be added. Please see Cubelli et al., 2017 and Rode et al., 2017 (See references for details).

Methods

I appreciate to report the specific question 11 in a box and the description of NIHSS. Is it possible to move the description of NIHSS in the line 74 after its first mention?.

Analysis

I understand the problem of significance (and effect size) with a very large numbers.

Discussion

Reference for gender difference discussion is McGlone, Losier, Black, 1997

References for discussing frequency are: Yue et al., 2012; Di Monaco et al., 2011; Buxbaum et at., 2004; Vallar et al., 1994; Stone, Halligan, Greenwood, 1993; Stone et al., 1992; Ogden, 1987.

Some discussion of these difference should be made (Test, days from stroke, criteria etc…)

Strengths and Limitations

Please add also the problem of hidden – personal neglect. See Caggiano et al., 2014 and Caggiano et al., 2018

References:

Cubelli, R. (2017). Definition: Spatial neglect. Cortex, 92, 320.

Rode, G., Pagliari, C., Huchon, L., Rossetti, Y., & Pisella, L. (2017). Semiology of neglect: an update. Annals of Physical and Rehabilitation Medicine, 60(3), 177-185.

Caggiano, P., Beschin, N., & Cocchini, G. (2014). Personal neglect following unilateral right and left brain damage. Procedia-Social and Behavioral Sciences, 140, 164-167.

Caggiano, P., & Jehkonen, M. (2018). The ‘neglected’personal neglect. Neuropsychology review, 28(4), 417-435.

Author Response

Reviewer 1

Author's Notes to Reviewer

Thank you for reviewing our manuscript and providing us with the opportunity to address your concerns and suggestions. We have responded below to your points and have altered sections of our paper in line with your recommendations. The changes from the original text are highlighted in the manuscript.

Comments and Suggestions for Authors

I read with interest this large scale - population based analysis. The paper is well written and reports an extensive analysis that confirm previous studies in a different form, analyses and population. The authors correctly conclude that the results are useful to take decision in future policies, statements and service provision. However there are some minor points to change to improve the readability of manuscript to a wider audience and include some recent points.

Title

Ok

Abstract

I think the “modified Rankin” is the “modified Rankin Scale.”. Please add this description.

We have rectified this error throughout.

Introduction

I agree with the description of spatial neglect, but a recent definition could be added. Please see Cubelli et al., 2017 and Rode et al., 2017 (See references for details).

Thank you for your suggestions of these more recent references. We have altered our definition and added these references. It now reads:

‘It is a syndrome rather than a single impairment [3], central to which is a cognitive disorder of attention and spatial awareness and information processing in which patients characteristically fail to orientate, report or respond to stimuli located on the contralesional side to the stroke [1, 4-5].’

Methods

I appreciate to report the specific question 11 in a box and the description of NIHSS. Is it possible to move the description of NIHSS in the line 74 after its first mention?.

We have changed the location of Box 1 in the text.

Analysis

I understand the problem of significance (and effect size) with a very large numbers.

Discussion

Reference for gender difference discussion is McGlone, Losier, Black, 1997

Thank you for pointing out the reference. We have altered the text and added the reference. It now reads:

’ An unexpected finding was that neglect was observed more frequently in women, as previous studies found the incidence to be equal across gender when performing drawing tasks [31].‘

References for discussing frequency are: Yue et al., 2012; Di Monaco et al., 2011; Buxbaum et at., 2004; Vallar et al., 1994; Stone, Halligan, Greenwood, 1993; Stone et al., 1992; Ogden, 1987

We have included some of these references to aid the discussion regarding prevalence. This now reads:

‘In line with this assumption, the prevalence of neglect varies between reports which appears to be associated with the assessment detail performed in the different studies [40-41].’

Some discussion of these difference should be made (Test, days from stroke, criteria etc…)

We have clarified that the tests used and their timing would affect diagnosis. The section now reads:

‘In line with this assumption, the prevalence of neglect varies between studies, depending on the choice and timing of test used, diagnostic criteria and different types of neglect [39-41].’

Strengths and Limitations

Please add also the problem of hidden – personal neglect. See Caggiano et al., 2014 and Caggiano et al., 2018

We have added personal neglect as an example of the atypical impairment we describe in the Limitations sections and the Caggiano et al 2018 reference. It now reads:

‘We recognise the simplification, limitation and imprecision of this measure and that it is possible that mild or atypical impairments, like personal neglect [36], may have been missed [37-38] with a likely underestimation of the prevalence of neglect [37-38].’

References:

Cubelli, R. (2017). Definition: Spatial neglect. Cortex, 92, 320.

Rode, G., Pagliari, C., Huchon, L., Rossetti, Y., & Pisella, L. (2017). Semiology of neglect: an update. Annals of Physical and Rehabilitation Medicine, 60(3), 177-185.

Caggiano, P., Beschin, N., & Cocchini, G. (2014). Personal neglect following unilateral right and left brain damage. Procedia-Social and Behavioral Sciences, 140, 164-167.

Caggiano, P., & Jehkonen, M. (2018). The ‘neglected’personal neglect. Neuropsychology review, 28(4), 417-435.

Reviewer 2 Report

This is a well conducted study on a very lage dataset that characterize the prevalence of neglect after hopsitalization for neurological reasons.

The overall picture that is provided to the reader is intriguing. I believe these data are very interesting for clinical and experimental neuropsychologists, as well as for health economists.

While I share the authors' view that with so large a sample statistical inference is of little use and would even be misleading, there is one point on which at least estimation - if not inference - would be needed.

Data look rather dramatic in terms of the consequences of (or associated outcomes) of neglect. However, we all know that neglect is very often associated with hemiplegia, and the latter is the most obvious predictor of poor outcome, need for rehabilitation, social costs, etc. So, I believe the authors shuould provide either separate tables for patients with and without hemiplegia, if not some finer co-variation statistical technique.

To explain the same point at a more general level, if one characterizes a large sample of patients for the prevalence of symptom X of stroke and for a number of associated variables (outcome, rehabilitation, etc.), the straightforward question arises whether it is not stroke per se that is associated to those variables, rather than symptom X. One feasible way to ascertain whether this is the case, even at the descriptive level the authors wish to keep their paper on, is to report whether another very important symptom Y, notoriosuly linked to social/health costs, "takes" the effect of X away, and to what degree. Y might well be hemiplegia.

Another point any expert in neglect would be happy to know about, is to what extent this large dataset reflects the notorious imbalance of neglect frequency/severity after right vs left hemisphere stroke. I believe such a datum would be of great interest to all neuropsychologists.

A minor point. The authors note in the Dicussion: "We found that neglect was more common in individuals who were dependent before their admission, but co-morbidities such as diabetes and previous stroke/TIA although present were curiously less frequent." Given the nature of the data, some form of selection bias by the examiners filling in the forms is likely here. If a patient has severe (current) deficits to be reported in the form, the examiner probably focused on them and relatively overlooked the patient's history. If the patient has just mild deficits, it is likely that the attention of the examiner falls more often on the patients' history. I believe this "attentional" bias by examiners is to be expected.

Author Response

Reviewer 2

Author's Reply to the Review Report (Reviewer 2)

Thank you for reviewing our manuscript and providing us with the opportunity to address your concerns and suggestions. We have responded below to your points and have altered sections of our paper in line with your recommendations. The changes from the original text are highlighted.

Comments and Suggestions for Authors

This is a well conducted study on a very large dataset that characterize the prevalence of neglect after hopsitalization for neurological reasons.

The overall picture that is provided to the reader is intriguing. I believe these data are very interesting for clinical and experimental neuropsychologists, as well as for health economists.

While I share the authors' view that with so large a sample statistical inference is of little use and would even be misleading, there is one point on which at least estimation - if not inference - would be needed.

Data look rather dramatic in terms of the consequences of (or associated outcomes) of neglect. However, we all know that neglect is very often associated with hemiplegia, and the latter is the most obvious predictor of poor outcome, need for rehabilitation, social costs, etc. So, I believe the authors shuould provide either separate tables for patients with and without hemiplegia, if not some finer co-variation statistical technique.

To explain the same point at a more general level, if one characterizes a large sample of patients for the prevalence of symptom X of stroke and for a number of associated variables (outcome, rehabilitation, etc.), the straightforward question arises whether it is not stroke per se that is associated to those variables, rather than symptom X. One feasible way to ascertain whether this is the case, even at the descriptive level the authors wish to keep their paper on, is to report whether another very important symptom Y, notoriously linked to social/health costs, "takes" the effect of X away, and to what degree. Y might well be hemiplegia.

We acknowledge that neglect is associated with hemiplegia as well as other impairments, e.g. hemianopia, as indicated by the increased severity of stroke in the NIHSS. All of these impairments could affect the outcome after stroke. We are very aware that confounding is possible when presenting large datasets and we therefore highlight that in our method section (page 3 row 105). “Due to the large sample size, it would be highly likely that any simple comparison (e.g. t-test) would be statistically significant regardless of the size of effect. The data structure and confounding present in this observational, records-based dataset would require complex methodology to produce robust conclusions. Given our objective to describe the sample population, these were not planned for a priori or reported here.”

Unfortunately for our purposes the SSNAP database does not record hemiplegia. We therefore cannot investigate this and other likely confounding but rather only describe the sample population and the associations with other outcomes. We have included below in the limitation section which now reads:

‘Therefore we cannot say that the poor outcomes observed were caused by the neglect and acknowledge that they could be attributable to another symptom, like hemiplegia. Due to the nature of the dataset and the descriptive analysis we performed, we at no point statistically analyse any associations with outcome measures, and in turn would not be able to identify if causal relationships are present. Any interpretation should therefore be done with great care.’  

Another point any expert in neglect would be happy to know about, is to what extent this large dataset reflects the notorious imbalance of neglect frequency/severity after right vs left hemisphere stroke. I believe such a datum would be of great interest to all neuropsychologists.

This would indeed be valuable to add but it is not possible to extract this variable with certainty from the SSNAP database. During our data analysis we attempted to extrapolate the affected body side using motor weakness of the arm and leg from the NIHSS. However, this did not result in a clear division of sides and a large proportion of individuals presented with no weakness, bilateral weakness or weakness of the arm and leg on opposite sides. This can probably be attributed to impairments other than hemiplegia, previous weakness etc. We concluded that this method was not accurate in establishing which hemisphere is affected and did not include the analysis in the paper.

We have now added the below to the limitations section in the paper:

‘For this size of database it would be of great interest to record the incidence of neglect in left and right hemisphere stroke. Unfortunately this is not information that is captured in the SSNAP database and therefore we can not report on this aspect.’

A minor point. The authors note in the Discussion: "We found that neglect was more common in individuals who were dependent before their admission, but co-morbidities such as diabetes and previous stroke/TIA although present were curiously less frequent." Given the nature of the data, some form of selection bias by the examiners filling in the forms is likely here. If a patient has severe (current) deficits to be reported in the form, the examiner probably focused on them and relatively overlooked the patient's history. If the patient has just mild deficits, it is likely that the attention of the examiner falls more often on the patients' history. I believe this "attentional" bias by examiners is to be expected.

This is difficult to ascertain and we therefore have included this point in the manuscript which now reads:

Alternatively, there may be selection bias in entering routinely collected clinical data into the national audit database with clinicians and audit clerks more focused on severe stroke related deficits than milder problems in the person’s medical history.

Round 2

Reviewer 2 Report

I seem to understand from the following website that NIHSS contains items (5A, 5B, 6A, 6B) which regard hemiplegia:

https://www.mdcalc.com/nih-stroke-scale-score-nihss

The authors explained in their reply that SSNAP does not contain those items - this is weird indeed, as they are probably the most relevant overall for rehabilitation programmes. The authors should overtly comment in the paper on this drawback of SSNAP, as it loses so obvious a landmark to understand the impact of stroke on daily life. I still cannot believe it lacks them!

While exploring NIHSS, another critical item emerged, i.e. the one concerning visual field (item 3): it is also very relevant to neuropsychologists to know about the general association between neglect and visual field loss. Hoping that SSNAP contains at least this item, a table with the cross-diagnosis of neglect and hemianopia would be great, and would add considerable interest to the report.

Of course, this is not to say that the currently reported data are not interesting, far from that - however I believe that access to that huge dataset grant a perhaps unique opportunity to extract more information, which in turn would give some stronger and more complete picture.
